Review  

evolution, microbiology, ecology

evolution, gut microbiome, diversity, ecological opportunity, colonization, bacteria

**Author for correspondence:**
Pauline D. Scanlan
e-mail: p.scanlan@ucc.ie

One contribution to the Special Feature 'Application of ecological and evolutionary theory to microbiome community dynamics across systems'. Guest edited by Dr James McDonald, Dr Britt Koskella and Professor Julian Marchesi.

# Microbial evolution and ecological opportunity in the gut environment

Pauline D. Scanlan[1,2]

[1]School of Microbiology, and [2]APC Microbiome Ireland, Biosciences Building, University College Cork, Cork, Ireland

PDS, 0000-0002-6733-9653

Recent genomic and metagenomic studies have highlighted the presence of rapidly evolving microbial populations in the human gut. However, despite the fundamental implications of this intuitive finding for both basic and applied gut microbiome research, very little is known about the mode, tempo and potential functional consequences of microbial evolution in the guts of individual human hosts over a lifetime. Here I assess the potential relevance of ecological opportunity to bacterial adaptation, colonization and persistence in the neonate and germ-free mammalian gut environment as well as over the course of an individual lifetime using data emerging from mouse models as well as human studies to provide examples where possible. I then briefly outline how the continued development and application of experimental evolution approaches coupled to genomic and metagenomic analysis is essential to disentangling drift from selection and identifying specific drivers of evolution in the gut microbiome within and between individual human hosts and populations.

## 1. Introduction

Much of gut microbiome research seeks to link microbial diversity with microbial function and some aspect of the human host phenotype (e.g. health or specific disease state). Microbial diversity can be conceptualized and studied at different levels of hierarchy, and the advent of high-throughput sequencing technologies, culturomics and bioinformatics means that we can now, in principle, study variation in diversity at different levels of resolution from the highest taxonomic levels all the way through to fine-grain analysis at the strain level within an individual for a considerable proportion of this microbial community [1–3]. However, until relatively recently the vast majority of gut microbiome studies that have tried to link microbial diversity and function to human phenotypes of interest have profiled bacterial diversity at the species or higher levels of taxonomic resolution (typically genus level due to limited genetic resolution of common biomarkers e.g. 16S rRNA coupled to short-read lengths) [4–6]. As a consequence, the extent of variation in microbial diversity within and between individuals is likely to be greatly underestimated. Moreover, this type of analysis masks the potential presence of dynamic and rapidly evolving sub-species populations of gut microbes including bacteria [7] and bacteriophages [8]. Given our knowledge of gut microbial populations' sizes, rapid generation times and estimated mutation rates the exciting idea that microbes are rapidly evolving within their human hosts over time-scales of weeks and months is intuitive [9,10]. However, it is only with the recent publication of different studies using whole genome sequence analysis of longitudinally sampled bacterial isolates [7,11,12] and/or high resolution metagenomic analysis of bacteria and bacteriophages [8,13] that this obvious but greatly overlooked reality is being confirmed.

Evolution is defined as genetic change in a population from one generation to the next. Genetic differences between individuals in microbial populations

arise from mutational events that can range from single nucleotide changes, insertions and deletions to recombination and horizontal gene transfer events. The rise and spread of novel genetic variants or alleles within a population from one generation to the next can then occur via random or non-random evolutionary processes. Random processes such as genetic drift result in the chance spread of selectively neutral or nearly neutral mutations in a population or in the case of genetic draft whereby mutations can hitchhike on a beneficial genetic background and increase in frequency [14–16]. Conversely, under natural selection mutations that confer a benefit (i.e. increase the fitness or reproductive success of an individual) can increase in a population due to the selective advantage they impart [17]. Although we know virtually nothing about the scale and extent of microbial evolution in the human gut ecosystem during the life of an individual and the relative importance of selection over drift, the ability of microbes to evolve in real-time is potentially paradigm shifting in terms of how we think about their functionality—the very simple notion that the phenotype of an organism can evolve to vary in time and space has obvious and manifold implications for both basic and applied gut microbiome research. From the broadest perspective, if we are to better understand microbiome diversity and function within and between human individuals and populations we require basic knowledge of the mode, tempo and potential drivers of microbial evolution within this ecosystem. Addressing this will undoubtedly require intensive research efforts, but information on microbial evolutionary processes in the human gut will allow us to better understand the relationship between genetic diversification and microbial function below the species level, help explain observed phenomena (e.g. the uniqueness of each gut microbiome in terms of microbial diversity) and inform the principles that govern the colonization success, persistence, stability and resilience of gut microbes and the gut microbiome. Moreover, if microbial function can evolve rapidly within and between individuals then how we think about and design experiments and gather and interpret data on the microbiome requires change. Trials and interventions that aspire to manipulate the microbiota in both health and disease states may be better planned and more successful by incorporating an evolutionary perspective.

Several studies have used (meta)genomic comparative approaches to show that adaptation to host diet has played a primary role in shaping the microbial diversity of the human gut over the course of human evolutionary history [18,19]. However, humans are long-lived hosts and, as recent publications indicate, within an individual's lifetime microbes that inhabit the gut are evolving over short timescales of days, weeks and months [7,8,11,12]. As such, it is conceivable that over a given human lifespan there is sufficient time and opportunity for new strains and species of microbes to evolve *de novo* within the human body. In fact, it has been shown that certain gut bacteriophages are diversifying at a rate that is equivalent to the evolution of a new phage species within the short time scale of 2 years [8]. Although there are a number of specific factors that can drive microbial evolution (e.g. parasitism [20]), for this perspective I will specifically focus on the potential relevance of ecological opportunity and its role as a driver of bacterial evolution and diversification in the gut ecosystem. I will then briefly outline how complementary analyses that marry experimental evolution and ecology approaches with the study of naturally occurring gut communities may provide the means to bridge the gap from inferring selective forces and evolutionary processes based on genetic, phenotypic and metadata alone to directly studying them in a controlled, replicated and tractable manner.

## 2. Ecological opportunity

Although definitions of ecological opportunity have differed slightly in the literature, a recent review has consolidated variations on the theme into a precise mechanistic definition [21]. Ecological opportunity is defined as a prospective, lineage specific characteristic of an environment that contains both niche availability that enables the persistence of the lineage within an environment *and* niche discordance which drives diversifying selection within that lineage [21]. Niche availability refers to the capacity for a lineage with a particular phenotype that is new to the community to persist within that community and niche discordance is what drives diversifying selection due to the discordance or adaptive mismatch between a lineage's niche-related traits and the new ecological conditions encountered by the lineage. Ecological opportunity can therefore arise from changes in the environment in the lineage's current location or it can result from the colonization of a new location by a lineage. Ecological opportunity encompasses community context and is subject to changes in both biotic and abiotic components of the environment. As such, ecological opportunity is a broad term and ecological opportunity can manifest in multiple forms (e.g. access to novel resources, extinction of a competitor species or predator, abiotic stress). Moreover, how a given lineage responds to ecological opportunity will depend on the diversification potential and the temporal and spatial scale of heterogeneity in ecological opportunity [21]. Of note, *in vitro* microcosm experiments and *in vivo* mouse models have shown that bacteria can diversify and adapt in newly colonized and heterogeneous environments over very short time scales (days) [22–25].

## 3. Ecological opportunity in pristine and low diversity gut environments

Given that individual humans are unique in terms of genetics and lifestyle (diet, medication, etc.) and show considerable variation in microbiota composition and species richness over a lifetime [26,27], ecological opportunity within the gut ecosystem is likely to vary in time and space within and between individuals in a population. At birth we can conceptualize the human gut of each individual as a pristine or at least depauperate ecosystem [28] that represents a unique adaptive landscape with considerable ecological opportunity and potential for adaptive radiation. However, the adaptive and diversification potential for colonizing microbes is likely to vary based on the specific microbe(s) in question, the host immune system and extrinsic factors relating to birth mode, infant diet and exposure to antibiotics. For example, the neonate gut can be seeded by microbes from a range of sources, including maternal (birth canal, faeces, skin, breast milk) and environmental (hospital setting, home environment, other humans) [29–31], but it may be

that niche discordance is less for vertically transmitted organisms whose lineages have been faithfully transmitted over generations between mother and offspring. Here, microbes that are acquired via passage through the birth canal, exposure to maternal faeces and/or breast milk may already be well adapted to the neonate gut environment and ecological selection favours their colonization, with perhaps few mutational events required for these microbes to adapt to the new host gut. Although we do not have any explicit studies of ecological opportunity in the human gut we can consider its relative importance in driving the diversification of well-recognized vertically transmitted and early colonizers of the infant gut such as *Bifidobacterium* spp. [32] compared with potentially less-well adapted horizontally (environmentally) acquired species. Many infant-associated *Bifidobacterium* species (e.g. *Bifidobacterium breve*, *Bifidobacterium bifidum*) can use specific glycans (human milk oligosaccharides) found in breast milk that cannot be metabolized by the host [33]. This metabolic capacity highlights their adaptedness to the breastfed infant gut and affords them a considerable ecological advantage over other potentially colonizing microbes that do not possess this trait and therefore may get outcompeted and cleared from the gut environment more readily in the absence of other suitable resources they can exploit. While the reduced time lag potentially afforded by vertical transmission of microbes from mother to infant may contribute to priority effects, we must also consider that successful establishment and colonization may then be enhanced via mutation and selection (i.e. adaptation to the new environment), which can in turn facilitate persistence within a new human host. A recent study that used a metagenomics based approach to track potential vertical transmission events between mother and infant pairs showed that even though certain strains of *Bifidobacterium* detected in the infant gut at three months post-partum were highly related to those in the mother and were likely to be vertically transmitted they had a degree of nucleotide divergence from the maternal strain [34]. These data may indicate that although this strain was successfully transmitted from mother to infant it continued to evolve even over the short period of three months. Another similar metagenomics-based study noted the persistence of maternally acquired strains in the infant gut that had also diverged from the maternal strain through time which the authors attributed to strain replacement or mutation [30].

Vertical transmission clearly plays an important role in colonization success and separate studies have shown that vertically transmitted strains acquired from the maternal gut persist for longer (over a four month period) in the infant gut compared to non-gut and horizontally acquired strains [35] and that the majority of vertically transmitted strains detected at high abundance in vaginally delivered babies at day four post-partum were still present 1 year later [36]. However, there may be situations where vertical transmission from mother to infant is negatively affected [30] (e.g. in infants that are born via C-section, are not breastfed and where antibiotics have been administered to mother and/or infant during or after delivery), which in turn could provide a greater opportunity for horizontally transmitted microbes to colonize. For example, a recent metagenomics study of 596 full-term babies highlighted the negative impact of C-section delivery and antibiotic prophylaxis on the maternal transmission of gut bacteria such as

pioneering *Bacteroides* species [29]. In such a scenario where the vertical transmission chain is negatively affected perhaps mutation and natural selection are crucial to successful colonization of the gut environment for horizontally acquired microbes. Moreover, if there is sufficient niche availability but higher niche discordance we might anticipate observing higher rates of molecular evolution in horizontally acquired microbes. In line with this reasoning one of the first studies of microbial evolution in the mammalian gut showed that elevated mutation rates confer a selective advantage when colonizing this environment [37]. In this study, Giraud and co-workers colonized (separately) germ-free mice with two variants of an *Escherichia coli* strain (K12 MG1655) that was isogenic except for mutation rate to explicitly look at how bacterial mutation rates impact on colonization success in the gut environment. The *E. coli* strain used was a laboratory strain and not isolated from the murine gut and therefore one could anticipate that niche discordance is certainly higher for this laboratory strain than a native murine *E. coli* strain. Intuitively, the mutator phenotype was found to be initially beneficial as it facilitated the supply of adaptive mutations required for successful colonization and the *de novo* evolution of higher bacterial mutation rates during colonization in the initially non-mutator strain was also observed in a small subset of the murine hosts (8%, $n = 26$). However, a higher mutation rate was no longer beneficial once adaptation was achieved and mutations accumulated in these bacteria during colonization were in fact detrimental in secondary environments. Of note, this phenotype has been reported in the recent study of microbial evolution in the adult gut [11]. This finding suggests that an increased mutation rate is not only beneficial to a bacterial lineage during the colonization of a new niche but may also be favoured by natural selection to facilitate adaptability in a lineages current environment.

Adaptive radiations are a recognized signature of ecological opportunity [21,38] and, similar to the aforementioned mouse model study, others have used strains of *E. coli* to explicitly investigate divergence and niche adaptation in the mammalian gut. One such study monitored genetic changes associated with *E. coli* K12 MG1655 colonization and diversification in the guts of germ-free mice [23,24]. Within a few days post-inoculation three distinct mutants, each with unique phenotypes relative to the wild-type and each other, emerged and stably coexisted over the time scale of the experiment. Mutations in global regulators and genes associated with motility, carbon source utilization as well as membrane transport and permeability in response to osmolarity were observed [23,24]. The pattern of diversification observed was typical of an adaptive radiation and a trade-off between resistance to bile salts and nutritional competency was proposed based on the genotypes of evolved bacteria together with the phenotypic changes in motility and bacterial growth (fitness) observed under different resource and stress conditions [24]. Another study that also used *E. coli* K12 MG1655 to investigate adaptation to the gut environment but in a different mouse model (streptomycin-treated mouse model that in theory and in practice maintains anaerobic bacteria in the gut) observed differences in the targets of selection and found mutations primarily in genes involved in galactitol and sorbitol metabolism as well as transmembrane transporters [39]. A more recent study that used a non-laboratory *E. coli* strain in the same type of

streptomycin-treated mouse model observed even further differences in the genes under selection compared to those observed in *in vitro* and in these aforementioned mouse models [40]. Here, a lower rate of adaptation compared with other *in vitro* and *in vivo* studies and no mutations in global regulators were observed. Instead, mutations in the D-galactonate operon (which facilitated growth on this carbon source *in vitro*) and mutations in genes associated with ribosome maturation were noted.

Collectively, these data emerging from different *E. coli*–mouse model evolution studies suggest that the timescales of the experiment as well as specific experimental details relating to the mouse model systems may contribute to differences in the data observed (e.g. in the latter study mutations in genes associated with ribosome maturation improve bacterial growth in the presence of streptomycin which was added to the mouse drinking water before colonization and throughout the experiment [40]). However, variation observed across these different studies also indicate that the genotype of the colonizing strain and the degree of niche discordance between it and the environment encountered is also of critical importance to understanding the likely targets of selection and the underlying genetics of adaptation associated with colonization of the gut environment. Interestingly, while these studies highlight the capacity for different *E. coli* strains to rapidly adapt to different models, a recent study of *E. coli* evolution in the human gut over several months did not find any signal of adaptation [12]. What is particularly intriguing about this contrasting finding is that it might tell us something about ecological opportunity, niche discordance and adaptive potential in the gut environment. Here, researchers sampled *E. coli* populations from an individual at three different time points over 315 days and sequenced 24 isolates of the dominant *E. coli* (ED1a) strain within the population. Analysis revealed very little genomic diversity with no evidence of selection. As outlined this is in stark contrast to what has been observed in many of the aforementioned experimental evolution studies using *E. coli*. However, given that the strain sequenced was dominant within the *E. coli* population of that individual this might indicate that perhaps this strain is well adapted to its host and this is why it appears to be evolving neutrally through time [12]. With respect to this finding the application of an experimental evolution approach using murine strains to colonize murine models may provide more meaningful insight into ecological opportunity and bacterial adaptation to the gut environment relative to what has been found in aforementioned mouse models studies that have used laboratory or human strains.

Finally, an understanding of how mutation and natural selection contributes to the relative success of individual microbes when colonizing a pristine or low diversity gut is also key to the study of community assembly. However, the degree to which adaptation can contribute to successful colonization and in turn priority effects, including niche pre-emption and niche-exclusion, and ultimately community assembly and the persistence of microbes in the human gut while recognized remains unknown [41]. Nonetheless, evolution-mediated priority effects, also known as monopolization, is increasingly recognized as important to the dynamics of community assembly [42] and has been demonstrated in experimental microbial populations undergoing diversifying selection [43].

## 4. Is ecological opportunity a constant selective force in the gut over the lifetime of an individual human host?

Ecological opportunity is perhaps easiest to conceptualize and study under germ-free and/or depauperate conditions, and is of clear relevance to our understanding of how microbial evolution impacts on the colonization and diversification of microbial species in the neonate and germ-free gut environment. It is also easy to consider how the spatially structured nature of the gut, which provides extensive variation in the physical and chemical environment, hosts a range of micro-niches that probably impacts on ecological opportunity. Such micro-niche variation runs both longitudinally from the oral cavity to the rectum and radially from the lumen to the host epithelium and can dictate the local biogeography or spatial organization and compositional variation of microbes at different sites in the gut both in health and disease [44–46]. However, we must also take a dynamic perspective and think about how humans are long-lived hosts and the physical, immunological and physiological changes our bodies undergo [26,47]. For example, normal development processes affecting the human body, including key transitions in life-stages (e.g. infancy, childhood, adolescence, adulthood and old-age), can all result in changes to gut surface area and heterogeneity, maturation and function of the host immune and endocrine systems, and microbiome complexity [47–50]. Variation in these factors through time will undoubtedly impose novel diversifying selection pressures on both extant and invading microbes, and therefore it is likely that there is both sufficient niche availability and niche discordance for ecological opportunity to perhaps serve as a constant, or at the very least a periodic selective force in the gut environment over an individual human lifetime.

In addition to natural physiological changes another factor that may affect ecological opportunity is the onset of a disease or infection that alters the gut environment. Many high-burden chronic diseases are associated with changes in both the biotic and abiotic environment of the gut including altered intestinal motility and immune responses, together with changes in microbial diversity [51]. Such changes are characteristic of many common gastrointestinal diseases including irritable bowel syndrome (IBS) [52] and inflammatory bowel diseases (IBDs) [53]. Moreover, individuals with diseases such as IBDs and IBS experience periods of remission and relapse such that there are potential large shifts in gut environmental variation over relatively short periods of time [54–56]. Again, the ecological changes associated with disease and disease onset on the microbiota are relatively well documented, and diseases such as IBD result in considerable changes in microbial species richness, stability and composition [57]. Although little is known about how microbial populations respond to changing environmental conditions associated with such diseases of the gut from an evolutionary perspective, within-host adaptation during disease progression has been well documented in other host organs such as the human lung. Here, the evolution of particular bacterial species such as *Pseudomonas aeruginosa* and *Burkholderia dolosa* in the lungs of cystic fibrosis (CF) patients has been studied and highlights the ability of microbes to adapt to both the disease state and interventions such as

antibiotic treatment [58,59]. For example, a retrospective study of a *B. dolosa* outbreak that sequenced isolates from 14 individuals over a 16-year period found signatures of parallel adaptive evolution in genes associated with bacterial membrane composition and antibiotic resistance, as well as oxygen dependent regulators which highlight the importance of mutation and selection in driving pathogen virulence within the human body [58]. In the case of *P. aeruginosa*, within-host population diversification in the CF lung results in a shift from an acute virulence to chronic host adapted phenotypes through time. Changes in phenotypes are facilitated by the evolution of mutator phenotypes, loss of function mutations in genes associated with quorum sensing together with selection for increased biofilm formation, mucoidy and antibiotic resistance [59]. With respect to pathogen evolution in the gut environment a recent study looked at the adaptation of a Crohn's disease (CD) associated pathogen to the gut environment during colonization and serial transmissions using adherent invasive *E. coli* (AIEC) in a streptomycin treated mouse model [60]. A diversity of AIEC mutants emerged with niche stratification, consistent with ecological opportunity, identified as the primary selective force. Different genetic variants emerged including one lineage that had enhanced acetate utilization relative to the ancestor and another with hypermotility that facilitated invasion and establishment in a mucosal niche [61]. A particularly informative aspect of the study was the analysis of 200 single *E. coli* isolates taken from 23 CD patient biopsy samples and 25 healthy controls as well as evolved commensal *E. coli* isolates (*E. coli* HS) that was used as a control and subjected to the same experimental regime as AIEC in the experimental model system. Consistent with the data from the model of AIEC adaptation to the gut, the collection of isolates obtained from CD patients, but not the healthy human controls and evolved *E. coli* HS isolates, were on average more motile and had a greater mean generation time when grown on acetate [61]. The strong correlation between the model and *in vivo* isolates underscores the biological relevance of the model as well as providing crucial insight into pathogen evolution and the differences in evolutionary trajectories of commensals and pathogens of the same species in the gut environment.

While such changes in human physiology and the biotic environment through time may serve as an important selective force perhaps externally derived factors may create the greatest divergent selection pressures on microbial populations residing in the human gut. Ecological opportunity is expected to increase during periods of rapid and varied environmental changes [21] and the radical changes in human ecology and lifestyle in recent decades are likely to have profound impacts on the ecological and evolutionary selection pressures that shape the diversity and function of the gut microbiota [62]. The ecological effects of a changing human lifestyle on gut microbiota composition have been documented and the microbiome of humans living in Westernized countries such as the US have a greatly reduced bacterial diversity compared with non-Western human populations [26,60,63]. However, the evolutionary effects of such changes on the gut microbiota remain unknown but disturbance events including exposure to lethal selection pressures such as antibiotics [64] and other medications [65], access to different resources through changes in diet and exposure to other xenobiotics [66–69] all have the potential to affect ecological opportunity by changing environmental and community context via the provision of novel resources and habitats to exploit, the removal or invasion of potential competitors or predators and parasites as well as the introduction of novel abiotic stressors.

Owing to their widespread use in human and agricultural applications and potentially lethal effects, antibiotics are probably the most common and severe selection pressure on bacterial populations [70]. The effects of antibiotic administration on gut microbiota diversity and composition has been extensively studied and disturbance events mediated by antibiotic administration [71] may result in conditions that support ecological opportunity within the gut environment. The emergence of antibiotic resistance strains is common and this has been frequently observed in individuals that receive large doses of antibiotics in clinical settings. In a recent study, researchers tracked the evolution of vancomycin-resistant *Enterococcus faecium* (VRE) populations in the gastrointestinal tract of patients undergoing allogeneic haematopoietic cell transplantation (allo-HCT) and in mouse models [7]. Genomic data from human and mouse model isolates demonstrated the capacity for rapid evolution of VRE and highlighted the presence of dynamic subpopulations of VRE that were diversifying on a daily basis. Many widely consumed prescribed and over-the-counter non-antimicrobial medications are also likely to affect ecological opportunity in the gut microbiome. A recent study that screened 38 common bacterial species against a diverse array of drugs that included 835 human-targeted drugs found that 203 or 24% inhibited the *in vitro* growth of at least one bacterial strain with 40 drugs affecting at least ten strains [65]. Similarly, a number of studies investigating the impact of different medications on the gut microbiome *in vivo* have consistently reported changes in bacterial diversity and relative abundances associated with non-antimicrobial medications such as metformin [72], non-steroidal anti-inflammatories [73] and laxatives [74].

Access to novel resources may also afford ecological opportunity and lead to evolutionary innovations mediated by horizontal gene transfer (HGT) as evidenced by a study on the origins of seaweed polysaccharide degrading enzymes in the gut microbiome of a subset of the Japanese population [75]. Seaweeds such as *Porphyra* constitute an important part of the Japanese diet and genomic analysis identified the presence of a porphyrin (specific type of carbohydrate in red seaweeds such as *Porphyra*) utilization locus of marine origin in the gut bacterium *Bacteroides plebeius* (strains of which have been isolated and sequenced from the intestinal microbiota of Japanese individuals). The presence of genes encoding for relaxase/mobilization functions which are required for conjugative DNA transfer downstream of the porphyrin utilization locus region identified a probable mechanism for HGT. Further comparative analysis of metagenomic datasets from Japanese ($n = 13$) and North American individuals ($n = 18$) identified seven potential porphyranases and six putative b-agarases (also involved in seaweed carbohydrate metabolism) in the microbiomes of four Japanese individuals with no porphyranase or agarase genes detected in the North American dataset [75]. Here, ingestion of marine seaweed with associated microbes is posited to have provided a potential novel niche (i.e. facilitated ecological opportunity) with evolutionary innovation mediated by HGT providing access to a novel resource in the gut by a resident microbe.

## 5. Linking gut microbiome research with experimental ecology and evolution

How we are born, what we eat, the medications we take, where we live and our interactions with the environment (as well as other people and animals) have changed considerably in recent years and decades. Consequently, both the biotic and abiotic adaptive landscape of the infant and adult human gut is likely to have changed considerably in recent times. How all these factors have and continue to affect ecological opportunity in the gut environment is an open question and disentangling the factors that drive genetic variation and separating drift from adaptation in the microbiome at the strain and species level is very challenging given the multitude of selection presences that a given human gut microbiome may be experiencing at any one time. To date, when looking to detect natural selection and adaptation in microbial populations, researchers typically look for signals of parallel evolution and convergence at varying different levels (e.g. gene, operon and pathway) and/or apply $K_a/K_s$ ratio analysis to longitudinal and/or cross-sectional genetic data. However, as alluded to, in the complex abiotic and biotic environment of the gut, individuals within populations may experience a multitude of different selection pressures at the one time or experience single different selection pressures at various different time-points. Moreover, ecological context plays a fundamental role in dictating evolutionary trajectories and as everyone has an individual host genotype, unique lifestyle and unique gut microbiota this automatically implies that the adaptive landscape for similar microbial strains and species is most likely to differ between the gut environments of individuals. Additionally, the use of non-synonymous to synonymous ratios may not always be appropriate as synonymous mutations have been found to have beneficial effects on fitness in experimental populations of microbes under strong selection [76,77]. Difficulties associated with trying to separate signal from noise when looking for signatures of adaptation in genetic data may be further compounded by clonal interference, epistasis, elevated mutation rates and horizontal gene transfer events, all of which have been observed in studies of microbial evolution in the gut environment [11,37,39,78]. Collectively, these different factors have the potential to mask signals of a selective force in operation for a given strain or species within an individual or subset of individuals within a larger group. As such researchers trying to identify the targets of selection and the specific and/or general factors driving adaptation within and between individuals as well as within and between human populations are often faced with an extremely difficult task.

However, progress is possible and is being made with our understanding of microbial evolution in the gut environment, greatly enhanced by the publication of the mouse model studies outlined throughout the text [23–25,37,39], as well as recent studies emerging from the genomic and metagenomics analysis of the gut microbiota of human populations [7,11–13,79]. The experimental evolve and re-sequence approaches coupled to the study of naturally occurring isolates are particularly powerful, as exemplified by the aforementioned study that used data derived from their mouse models studies to direct the phenotypic analysis of naturally occurring bacteria from the CD gut. This complementary analysis revealed phenotypic convergence with selection on traits associated with motility and acetate were

beneficial in a pathogenic genetic background in the gut environment [61]. However, in addition to mouse models, which afford the advantage to study the mode and tempo of microbial evolution in a physiologically complex and biologically relevant gut environment, another possible solution to better understand how microbes respond to selection and bridge the gap between the study of single strains *in vivo* is the use of chemostat or serial transfer 'microcosm' approaches [80–82].

Using chemostats or microcosms one could use faecal samples, focal species embedded in a complex community (e.g. faecal sample) or synthetic communities of microbes composed of diverse species of interest and representative of naturally occurring communities as inocula depending on the question and/or microbe(s) of interest at hand. Despite their lack of spatial structure and biotic complexity (e.g. lack of host immune system, etc.), these models are powerful experimental approaches that are used extensively in the field of microbial evolution owing to their experimental control, tractability and power of replication [82]. Moreover, experiments can be run indefinitely, and experimental conditions are potentially infinite, and can be chosen and manipulated in time and space according to the researcher's experimental design. With the lack of host backdrop caveat in mind, chemostats and microcosms may be particularly useful to investigate how different abiotic factors, such as antibiotics and resource availability, affect ecological opportunity and impact on microbiome evolution, ecology and function. Chemostat and microcosm experiments may also be readily applied to the study of how specific biotic interactions such as competition, parasitism (bacteria and bacteriophages) and predation (protists or bacterivores and bacteria) can promote or constrain ecological opportunity. While the data derived from *in vitro* experiments will be invaluable in itself, new hypotheses generated can then be tested in more complex models or can be used to parse metagenomics datasets for the presence or absence of the genetic signatures of adaptation uncovered in the experiments. Moreover, recent technological advancements have seen the development of an 'intestine-on-a-chip' that allows for the co-culture of both individual gut bacterial species and complex microbial communities containing diverse gut bacteria (including anaerobic and aerobic species) with mucous producing human villus intestinal epithelia [83]. Although co-culture experiments were run over short time-periods (between 3 and 5 days) it was indicated that co-culture experiments could be run for longer [83]. Crucially, the power of this model lies in its potential capacity to enable controlled replicated experiments to be conducted in an environment that recapitulates many important aspects of the physiology of the gut microbiome-host interface which is not possible using chemostats or microcosm approaches.

As is apparent from the text, *E. coli* is the most widely used model bacterium to study bacterial evolution in the gut. The reasons are manifold and linked to its ecological and clinical relevance together with ease of experimental and genetic manipulation and tractability. However, further progress into understanding microbial evolution in the gut requires us to move beyond focusing on this particular species and look at evolution in real-time across a broader range of species. As noted earlier, while rapid adaptive evolution of *E. coli* is common *in vitro* and in mouse models no signature of adaptive evolution was apparent for a particular

strain of *E. coli* in the adult human gut [12]. By contrast the recent work of Zhao *et al*. [11] showcased ongoing adaptive evolution of a taxonomically distant microbe—*Bacteroides fragilis*—in the complex environment of the adult gut over a 2-year period. This study combined the isolation and sequencing of *B. fragilis* strains longitudinally sampled from individual adults coupled to phenotypic assays and meta-genomic analysis of human population datasets from different geographical regions [11]. Of note, they found evidence of sub-ject specific adaptation related to polysaccharide import/binding but also signatures of adaptation across subjects in genes associated with cell envelope biosynthesis. With respect to cell envelope biosynthesis the authors speculated that the host immune system or evasion of phage predation may be the selective forces driving these adaptations in the gut. Consist-ent with the latter hypothesis, an experimental evolution study that looked at the genomic basis of bacterial adaptation to phages during coevolution detected multiple different non-synonymous mutations in genes associated with cell envelope biogenesis but none in their control populations that were evolved in the absence of phages [84].

As outlined, researchers working on the gut microbiome typically take a static perspective where microbial (mainly bacteria) species are fixed entities with specific functional traits. While certain functional traits may be fixed at the species and higher taxonomic levels there may be many important functional traits that impact on the host phenotype that are rapidly evolving within and between individuals. An understanding of what traits are open to and under selection is key to our understanding of the role of evolution in gut microbiota function. The key questions we need to address in future studies should centre on how ongoing microbial evolution at the sub-species level scales up to higher levels of microbial interactions and community structure and func-tion in the context of host health. For example, does microbial evolution buffer against environmental change and facilitate functional redundancy? What factors have the potential to drive or constrain within-host pathogenicity or other clini-cally relevant phenotypes (e.g. antibiotic resistance)? The degree to which microorganisms are locally adapted is also of fundamental importance to understanding community composition and stability and it may be that strains coloniz-ing the human gut are typically well adapted to gross features of the gut environment that are generally common to all hosts but microbes must continuously evolve to specifi-cally fine-tune adaptations to their current host or changes in the host environment to avoid going extinct or being replaced by a strain that is fitter. On this note, of particular importance is understanding how evolution plays out over the short term (weeks and months) versus over the longer term (years and decades) in the gut (i.e. selection on specific genes and traits over the short term may be costly in the long term if conditions changes such that strains may eventually become lost or replaced within the host or purifying selection may operate over the long term). Interestingly, perhaps the first study that capitalized on the availability of metage-nomics data revealed signatures of drift and purifying selection as the dominant evolutionary forces in a study of a subset of the human population [79]. More recently Garud *et al*. [13] used a modelling based approach rooted in population genetic theory to investigate evolutionary dynamics in the gut microbiome using longitudinal meta-genomic data from randomly sampled individuals within the American population, as well as twins. Analysis focused on 40 dominant bacterial species and although signatures of strong positive selection for some microbial species within individuals was evident in the short term the authors noted that strain replacement was most likely in the long term [13].

To date, the limited knowledge on microbial evolution in the gut may be due, in part, to a disconnect between disci-plines (i.e. gut microbiome research and evolutionary biology) whereby microbiome researchers are largely unaware of the capacity for microbes to evolve in real-time and have overlooked the potential importance of evolution to their research. However, the study of evolution in complex commu-nities is not trivial and technological roadblocks (e.g. low-cost, high-throughput sequencing is only available in the last decade) may have stymied progress. Nonetheless, even though the study of microbial evolution in the gut environ-ment is only in its infancy recent publications are paving the way. Importantly, these studies highlight the capacity for microbes to evolve within the body over very short time scales and help us conceptualize each human gut as a complex and varied adaptive landscape in which microbial evolution occurs. To further progress we must continue to move towards linking knowledge of evolutionary phenomena with applied research if we are to fully understand the implications of microbial evolution for gut ecosystem community structure and function in the context of host health.

Data accessibility. This article has no additional data.

Competing interests. I declare I have no competing interests.

Funding. My research is funded by a Royal Society–Science Foundation Ireland University Research Fellowship.

Acknowledgements. Thank you to the editors for the kind invitation to contribute to this special addition and to three anonymous reviewers for their excellent feedback.

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
