## [Reviewer comments · Proceedings of the Royal Society B: Biological Sciences]

Review History

RSPB-2019-1964.R0 (Original submission)

Review form: Reviewer 1 (Carolina Tropini)

Recommendation

Accept with minor revision (please list in comments)

Scientific importance: Is the manuscript an original and important contribution to its field?

Excellent

General interest: Is the paper of sufficient general interest?

Good

Quality of the paper: Is the overall quality of the paper suitable?

Excellent

Is the length of the paper justified?

Yes

Should the paper be seen by a specialist statistical reviewer?

No

Do you have any concerns about statistical analyses in this paper? If so, please specify them explicitly in your report.

No

It is a condition of publication that authors make their supporting data, code and materials available - either as supplementary material or hosted in an external repository. Please rate, if applicable, the supporting data on the following criteria.

Is it accessible?

N/A

Is it clear?

N/A

Is it adequate?

N/A

Do you have any ethical concerns with this paper?

No

Comments to the Author

I very much enjoyed reading this review, and found it struck a good balance between detailed examples and big picture ideas. In my mind, there are a couple other ideas that would be important to add in the context of microbial evolution and biological opportunity.

1. A brief discussion of biogeography, and how biotic and abiotic niches within the gut provide opportunities for evolution and ecological opportunity. As an addition to this, sequencing the stool provides only a time and space-averaged perspective.
 - Mark Welch, J. L., Rossetti, B. J., Rieken, C. W., Dewhirst, F. E. & Borisy, G. G. Biogeography of a human oral microbiome at the micron scale. *Proc. Natl. Acad. Sci.* 201522149 (2016). doi:10.1073/pnas.1522149113
 - Lavelle, A., Lennon, G., Winter, D. C. & O'Connell, P. R. Colonic biogeography in health and ulcerative colitis. *Gut Microbes* 7, 435–442 (2016).
 - Tropini, C., Earle, K. A., Huang, K. C. & Sonnenburg, J. L. The Gut Microbiome: Connecting Spatial Organization to Function. *Cell Host and Microbe* 21, 433–442 (2017).
 - Donaldson, G. P., Lee, S. M. & Mazmanian, S. K. Gut biogeography of the bacterial microbiota. *Nat. Rev. Microbiol.* 14, 20–32 (2016).
2. A brief discussion of how metagenomics can also be used for determining factors that are essential in a quantitative understanding of evolution, such as determination of generation time through techniques such as PTR or iRep.
 - Brown, C. T., Olm, M. R., Thomas, B. C. & Banfield, J. F. Measurement of bacterial replication rates in microbial communities. *Nat. Publ. Gr.* (2016). doi:10.1038/nbt.3704
 - Korem, T., Zeevi, D., Suez, J., Weinberger, A., Avnit-Sagi, T., Pompan-Lotan, M., Matot, E., Jona, G., Harmelin, A., Cohen, N., Sirota-Madi, A., Thaiss, C. A., Pevsner-Fischer, M., Sorek, R., Xavier, R. J., Elinav, E. & Segal, E. Growth dynamics of gut microbiota in health and disease inferred from single metagenomic samples. *Science* (80-.). 349, 1101–1106 (2015).

Minor comments:

Line 73: It is worth also noting that physical sampling may also masks the real bacterial diversity

due to extraction of DNA from a few hundred micrograms of material from a total of hundreds of grams, which may miss bacteria coming from rare biogeographical niches such as crypts, tight mucus etc.

Line 170: I think here it should be highlighted that environmental exposure plays a major role in seeding the neonatal microbiota, and that only a few vertically transmitted species appeared to be maintained over the long term.

- Korpela, K., Costea, P., Coelho, L. P., Kandels-Lewis, S., Willemsen, G., Boomsma, D. I., Segata, N. & Bork, P. Selective maternal seeding and environment shape the human gut microbiome. *Genome Res.* (2018). doi:10.1101/gr.233940.117

- Donald, B. and McCoy, K. D. Maternal microbiota in pregnancy and early life. *Science.* (2019). doi: 10.1126/science.aay0618

Line 380: Over the counter drugs are likely to have as big of an impact as antibiotics: antacids, laxatives, anti-inflammatory drugs are wildly prevalent and will heavily impact the gut environment.

- Maier, L., Pruteanu, M., Kuhn, M., Zeller, G., Telzerow, A., Anderson, E. E., Brochado, A. R., Fernandez, K. C., Dose, H., Mori, H., Patil, K. R., Bork, P. & Typas, A. Extensive impact of non-antibiotic drugs on human gut bacteria. *Nature* (2018). doi:10.1038/nature25979

- Tropini, C., Moss, E. L., Merrill, B. D., Ng, K. M., Higginbottom, S. K., Casavant, E. P., Gonzalez, C. G., Fremin, B., Bouley, D. M., Elias, J. E., Bhatt, A. S., Huang, K. C. & Sonnenburg, J. L. Transient Osmotic Perturbation Causes Long-Term Alteration to the Gut Microbiota. *Cell* 173, 1742–1754.e17 (2018).

- Imhann, F., Bonder, M. J., Vila, A. V., Fu, J., Mujagic, Z., Vork, L., Tigchelaar, E. F., Jankipersadsing, S. A., Cenit, M. C., Harmsen, H. J. M., Dijkstra, G., Franke, L., Xavier, R. J., Jonkers, D., Wijmenga, C., Weersma, R. K. & Zhernakova, A. Proton pump inhibitors affect the gut microbiome. *Gut* (2016). doi:10.1136/gutjnl-2015-310376

Line 476: A microcosm could also be encapsulated by a gut on a chip setup.

- Jalili-Firoozinezhad, S., Gazzaniga, F. S., Calamari, E. L., Camacho, D. M., Fadel, C. W., Bein, A., Swenor, B., Nestor, B., Cronce, M. J., Tovaglieri, A., Levy, O., Gregory, K. E., Breault, D. T., Cabral, J. M. S., Kasper, D. L., Novak, R. & Ingber, D. E. A complex human gut microbiome cultured in an anaerobic intestine-on-a-chip. *Nat. Biomed. Eng.* 3, 520–531 (2019).

Review form: Reviewer 2

Recommendation

Major revision is needed (please make suggestions in comments)

Scientific importance: Is the manuscript an original and important contribution to its field?

Good

General interest: Is the paper of sufficient general interest?

Good

Quality of the paper: Is the overall quality of the paper suitable?

Acceptable

Is the length of the paper justified?

Yes

Should the paper be seen by a specialist statistical reviewer?

No

Do you have any concerns about statistical analyses in this paper? If so, please specify them explicitly in your report.

No

It is a condition of publication that authors make their supporting data, code and materials available - either as supplementary material or hosted in an external repository. Please rate, if applicable, the supporting data on the following criteria.

Is it accessible?

N/A

Is it clear?

No

Is it adequate?

N/A

Do you have any ethical concerns with this paper?

No

Comments to the Author

The author provides a thoughtful perspective on current research in the gut microbiome field and how current and future work may benefit from a more thorough consideration of ecological and evolutionary principles. This manuscript highlights opportunities to unite ecological/evolutionary theory, microbiome science, and bacterial genetics and is likely to be of interest to investigators who take all three of these foci in their work. This is an important perspective that should be adopted broadly in the microbiome field.

Broadly, this article is well written and it highlights many key studies from an eco/evo vantage point. However, I feel that the manuscript would benefit from several revisions. These suggested revisions are highlighted below and mainly relate to delineating interpretations made by the authors of the primary literature versus interpretations made by the author of the review. Additional concerns relating to word choice, organization, and figure layout are also discussed.

Major comments

1. Line 194-196: The conclusion "...suggests that although this strain was successfully transmitted from mother to infant it continued to evolve even over the short period of three months" was not a key conclusion from Asnicar et al. (2017) mSystems. Though the authors do show data that demonstrates some divergence, this is an interpretation of the author of the manuscript in review. Clearer delineations throughout the manuscript of interpretations from the author and those already made in the primary literature will be important. This re-framing will be essential, in my opinion, to support the author's assertion from lines 551-553 that "microbiome researchers are largely unaware of the capacity for microbes to evolve in real-time and have overlooked the potential importance of evolution to their research."
2. The substance of the article speaks for itself and manuscript's only figure doesn't positively impact conclusions drawn from the text (even for a visual learner like myself). The author could consider reorganizing this figure into multiple figures that follow the text more closely, building a more informative figure, or including no figure at all.
3. Lines 281-285 and lines 496-499: This is a reasonable interpretation but an alternative

interpretation is that there are differences observed in the mouse experiments because the mice were inoculated with an exogenous strain rather than what was done in the human studies, where an already entrenched strain were used. If there were a fundamental difference in evolution and ecological opportunity between “mouse *E. coli*” versus “human *E. coli*,” an experiment observing native *E. coli* in mice or a non-native *E. coli* inoculated into humans would be essential.

4. Lines 369, 422: In addition to recent decades, the author should consider a brief discussion of changes to the human microbiome over longer time scales to frame the novelty of their viewpoint that evolution and ecological opportunity are operating at more recent timescales.

5. Lines 549-561: I feel that the author overlooked an important opportunity to discuss and conceptual roadblocks that existed previously (e.g. a decade ago) which do not exist now. Once you have the theory in mind, how do you leverage technology to design the appropriate experiment? We can't fault previous work if the methods did not exist to suitably address a hypothesis. That is, awareness of evolutionary theory is only the beginning and technological hurdles, rather than ignorance, may have fueled the microbiome field's lack of incorporation of eco/evo.

Minor comments

1. Line 70: Please be more specific “bacterial diversity at the species level or higher” should be “...at the species level or at higher taxonomic resolution.”

2. Line 90: “drift” not “draft”

3. Line 156: It may be helpful to the reader to highlight longer time scales after the shorter time scales are discussed (e.g. the text from later in the review [lines 329-342] about work done on adaptive radiation in the CF lung).

4. Lines 156-158: Please provide references.

5. Line 181: “Inaccessible to the host,” consider re-wording to “inaccessible to host metabolism.”

6. Line 184: “Get filtered out more readily” is imprecise. Consider changing to “excluded from the gut” or “outcompeted by other microbes and cleared from the gut environment.”

7. Line 193: “they had a degree of nucleotide divergence.” Was this degree of divergence significant?

8. Line 236-239. What were the strains that emerged?

9. Line 245: Please refer to K12 MG1655 as in lines 214 and 235. Or choose another consistent strain name in all cases.

10. Lines 299-302: Run-on sentence.

11. Line 303: The body also undergoes immunological changes throughout an individual's lifetime.

12. Lines 455, 457: Please provide re-reference key mouse and human studies.

13. Lines 463-468: Run-on sentence.

14. Lines 490-491: This sentence “As is apparent...” makes it sound like it's the best model to study bacterial evolution in the gut. *E. coli* is a minor player in the human gut microbiome in terms of abundance but has been utilized extensively in molecular biology, microbiology, etc. because of its tractability and relevance to some human diseases. Considering highlighting that *E. coli* isn't “the” model but “a” model.

15. Line 533: Please consider changing “superior” to “more fit”

16. Lines 534, 536: Consider mentioning days, weeks, months, decades, centuries, millennia and how these time scales are currently considered (or not considered) by the microbiome field.

Review form: Reviewer 3

Recommendation

Accept with minor revision (please list in comments)

Scientific importance: Is the manuscript an original and important contribution to its field?

Excellent

General interest: Is the paper of sufficient general interest?

Excellent

Quality of the paper: Is the overall quality of the paper suitable?

Excellent

Is the length of the paper justified?

Yes

Should the paper be seen by a specialist statistical reviewer?

No

Do you have any concerns about statistical analyses in this paper? If so, please specify them explicitly in your report.

No

It is a condition of publication that authors make their supporting data, code and materials available - either as supplementary material or hosted in an external repository. Please rate, if applicable, the supporting data on the following criteria.

Is it accessible?

N/A

Is it clear?

N/A

Is it adequate?

N/A

Do you have any ethical concerns with this paper?

No

Comments to the Author

This is a very nice and well written review on gut microbial evolution with a focus on discussing the importance of ecological opportunity. The author carefully cites most of the recent and older literature relevant to the subject and provides new ways forward.

I very much appreciated reading this paper.

I have a couple of really minor points that the author may consider.

line 93 can increase, instead of increase can

line 124 correct referencing.

line 136-154 if possible, shorten and give a more precise, and thus more testable, definition.

line 247: streptomycin treat lab mice in theory and in practice maintain a lot of anaerobic species.

line 447: consider adding evidence from mice provided in Frazão, Sousa, Lassig, Gordo 2019

PNAS.

Decision letter (RSPB-2019-1964.R0)

25-Sep-2019

Dear Dr Scanlan:

Your manuscript has now been peer reviewed and the reviews have been assessed by an Associate Editor. The reviewers' comments (not including confidential comments to the Editor) and the comments from the Associate Editor are included at the end of this email for your reference. As you will see, the reviewers and the Associate Editor have raised some issues with your manuscript and we would like to invite you to revise your manuscript to address them.

Research ethics:

Use of animals and field studies:

It is a condition of publication that you make available the data and research materials supporting the results in the article. Datasets should be deposited in an appropriate publicly available repository and details of the associated accession number, link or DOI to the datasets must be included in the Data Accessibility section of the article

(<https://royalsociety.org/journals/ethics-policies/data-sharing-mining/>). Reference(s) to datasets should also be included in the reference list of the article with DOIs (where available).

Please submit a copy of your revised paper within three weeks. If we do not hear from you within this time your manuscript will be rejected. If you are unable to meet this deadline please let us know as soon as possible, as we may be able to grant a short extension.

Best wishes,
Professor Hans Heesterbeek
mailto: proceedingsb@royalsociety.org

Associate Editor

Comments to Author:

Thank you for submitting your work for consideration as part of the special issue. Your paper has now been reviewed by myself and three reviewers, and we are all enthusiastic about the manuscript and think it will make an excellent contribution to the literature. That said, all three reviewers have made very thoughtful suggestions as to how the clarity of the work could be improved, and I would ask that you take each of these into account when revising your manuscript. In particular, reviewer 1 has made some great suggestions of additional concepts/ideas to cover, even if briefly, and suggested some additional citations. Reviewer 2 has asked that the author be more careful with their use of terms so as not to muddy the (already quite murky) waters in the field. They also suggest that the figure could be more informative if broken into pieces and matched more specifically with the structure of the text, and asks for

further discussion of alternative hypotheses in a few spots. Finally, reviewer 3 offers some minor suggestions for improvement. I look forward to reading your revised work, and thank you again for your great contribution.

Reviewer(s)' Comments to Author:

Referee: 1

Comments to the Author(s)

I very much enjoyed reading this review, and found it struck a good balance between detailed examples and big picture ideas. In my mind, there are a couple other ideas that would be important to add in the context of microbial evolution and biological opportunity.

1. A brief discussion of biogeography, and how biotic and abiotic niches within the gut provide opportunities for evolution and ecological opportunity. As an addition to this, sequencing the stool provides only a time and space-averaged perspective.

- Mark Welch, J. L., Rossetti, B. J., Rieken, C. W., Dewhirst, F. E. & Borisy, G. G. Biogeography of a human oral microbiome at the micron scale. *Proc. Natl. Acad. Sci.* 201522149 (2016).

doi:10.1073/pnas.1522149113

- Lavelle, A., Lennon, G., Winter, D. C. & O'Connell, P. R. Colonic biogeography in health and ulcerative colitis. *Gut Microbes* 7, 435–442 (2016).

- Tropini, C., Earle, K. A., Huang, K. C. & Sonnenburg, J. L. The Gut Microbiome: Connecting Spatial Organization to Function. *Cell Host and Microbe* 21, 433–442 (2017).

- Donaldson, G. P., Lee, S. M. & Mazmanian, S. K. Gut biogeography of the bacterial microbiota. *Nat. Rev. Microbiol.* 14, 20–32 (2016).

2. A brief discussion of how metagenomics can also be used for determining factors that are essential in a quantitative understanding of evolution, such as determination of generation time through techniques such as PTR or iRep.

- Brown, C. T., Olm, M. R., Thomas, B. C. & Banfield, J. F. Measurement of bacterial replication rates in microbial communities. *Nat. Publ. Gr.* (2016). doi:10.1038/nbt.3704

- Korem, T., Zeevi, D., Suez, J., Weinberger, A., Avnit-Sagi, T., Pompan-Lotan, M., Matot, E., Jona, G., Harmelin, A., Cohen, N., Sirota-Madi, A., Thaïss, C. A., Pevsner-Fischer, M., Sorek, R., Xavier, R. J., Elinav, E. & Segal, E. Growth dynamics of gut microbiota in health and disease inferred from single metagenomic samples. *Science* (80-.). 349, 1101–1106 (2015).

Minor comments:

Line 73: It is worth also noting that physical sampling may also masks the real bacterial diversity due to extraction of DNA from a few hundred micrograms of material from a total of hundreds of grams, which may miss bacteria coming from rare biogeographical niches such as crypts, tight mucus etc.

Line 170: I think here it should be highlighted that environmental exposure plays a major role in seeding the neonatal microbiota, and that only a few vertically transmitted species appeared to be maintained over the long term.

- Korpela, K., Costea, P., Coelho, L. P., Kandels-Lewis, S., Willemsen, G., Boomsma, D. I., Segata, N. & Bork, P. Selective maternal seeding and environment shape the human gut microbiome. *Genome Res.* (2018). doi:10.1101/gr.233940.117

- Donald, B. and McCoy, K. D. Maternal microbiota in pregnancy and early life. *Science.* (2019). doi: 10.1126/science.aay0618

Line 380: Over the counter drugs are likely to have as big of an impact as antibiotics: antacids, laxatives, anti-inflammatory drugs are wildly prevalent and will heavily impact the gut environment.

- Maier, L., Pruteanu, M., Kuhn, M., Zeller, G., Telzerow, A., Anderson, E. E., Brochado, A. R., Fernandez, K. C., Dose, H., Mori, H., Patil, K. R., Bork, P. & Typas, A. Extensive impact of non-antibiotic drugs on human gut bacteria. *Nature* (2018). doi:10.1038/nature25979

- Tropini, C., Moss, E. L., Merrill, B. D., Ng, K. M., Higginbottom, S. K., Casavant, E. P., Gonzalez, C. G., Fremin, B., Bouley, D. M., Elias, J. E., Bhatt, A. S., Huang, K. C. & Sonnenburg, J. L. Transient Osmotic Perturbation Causes Long-Term Alteration to the Gut Microbiota. *Cell* 173, 1742–1754.e17 (2018).

- Imhann, F., Bonder, M. J., Vila, A. V., Fu, J., Mujagic, Z., Vork, L., Tigchelaar, E. F., Jankipersadsing, S. A., Cenit, M. C., Harmsen, H. J. M., Dijkstra, G., Franke, L., Xavier, R. J., Jonkers, D., Wijmenga, C., Weersma, R. K. & Zhernakova, A. Proton pump inhibitors affect the gut microbiome. *Gut* (2016). doi:10.1136/gutjnl-2015-310376

Line 476: A microcosm could also be encapsulated by a gut on a chip setup.

- Jalili-Firoozinezhad, S., Gazzaniga, F. S., Calamari, E. L., Camacho, D. M., Fadel, C. W., Bein, A., Swenor, B., Nestor, B., Cronce, M. J., Tovaglieri, A., Levy, O., Gregory, K. E., Breault, D. T., Cabral, J. M. S., Kasper, D. L., Novak, R. & Ingber, D. E. A complex human gut microbiome cultured in an anaerobic intestine-on-a-chip. *Nat. Biomed. Eng.* 3, 520–531 (2019).

Referee: 2

Comments to the Author(s)

The author provides a thoughtful perspective on current research in the gut microbiome field and how current and future work may benefit from a more thorough consideration of ecological and evolutionary principles. This manuscript highlights opportunities to unite ecological/evolutionary theory, microbiome science, and bacterial genetics and is likely to be of interest to investigators who take all three of these foci in their work. This is an important perspective that should be adopted broadly in the microbiome field.

Broadly, this article is well written and it highlights many key studies from an eco/evo vantage point. However, I feel that the manuscript would benefit from several revisions. These suggested revisions are highlighted below and mainly relate to delineating interpretations made by the authors of the primary literature versus interpretations made by the author of the review. Additional concerns relating to word choice, organization, and figure layout are also discussed.

Major comments

1. Line 194-196: The conclusion “...suggests that although this strain was successfully transmitted from mother to infant it continued to evolve even over the short period of three months” was not a key conclusion from Asnicar et al. (2017) *mSystems*. Though the authors do show data that demonstrates some divergence, this is an interpretation of the author of the manuscript in review. Clearer delineations throughout the manuscript of interpretations from the author and those already made in the primary literature will be important. This re-framing will be essential, in my opinion, to support the author’s assertion from lines 551-553 that “microbiome researchers are largely unaware of the capacity for microbes to evolve in real-time and have overlooked the potential importance of evolution to their research.”

2. The substance of the article speaks for itself and manuscript’s only figure doesn’t positively impact conclusions drawn from the text (even for a visual learner like myself). The author could

consider reorganizing this figure into multiple figures that follow the text more closely, building a more informative figure, or including no figure at all.

3. Lines 281-285 and lines 496-499: This is a reasonable interpretation but an alternative interpretation is that there are differences observed in the mouse experiments because the mice were inoculated with an exogenous strain rather than what was done in the human studies, where an already entrenched strain were used. If there were a fundamental difference in evolution and ecological opportunity between “mouse *E. coli*” versus “human *E. coli*,” an experiment observing native *E. coli* in mice or a non-native *E. coli* inoculated into humans would be essential.

4. Lines 369, 422: In addition to recent decades, the author should consider a brief discussion of changes to the human microbiome over longer time scales to frame the novelty of their viewpoint that evolution and ecological opportunity are operating at more recent timescales.

5. Lines 549-561: I feel that the author overlooked an important opportunity to discuss and conceptual roadblocks that existed previously (e.g. a decade ago) which do not exist now. Once you have the theory in mind, how do you leverage technology to design the appropriate experiment? We can't fault previous work if the methods did not exist to suitably address a hypothesis. That is, awareness of evolutionary theory is only the beginning and technological hurdles, rather than ignorance, may have fueled the microbiome field's lack of incorporation of eco/evo.

Minor comments

1. Line 70: Please be more specific “bacterial diversity at the species level or higher” should be “...at the species level or at higher taxonomic resolution.”

2. Line 90: “drift” not “draft”

3. Line 156: It may be helpful to the reader to highlight longer time scales after the shorter time scales are discussed (e.g. the text from later in the review [lines 329-342] about work done on adaptive radiation in the CF lung).

4. Lines 156-158: Please provide references.

5. Line 181: “Inaccessible to the host,” consider re-wording to “inaccessible to host metabolism.”

6. Line 184: “Get filtered out more readily” is imprecise. Consider changing to “excluded from the gut” or “outcompeted by other microbes and cleared from the gut environment.”

7. Line 193: “they had a degree of nucleotide divergence.” Was this degree of divergence significant?

8. Line 236-239. What were the strains that emerged?

9. Line 245: Please refer to K12 MG1655 as in lines 214 and 235. Or choose another consistent strain name in all cases.

10. Lines 299-302: Run-on sentence.

11. Line 303: The body also undergoes immunological changes throughout an individual's lifetime.

12. Lines 455, 457: Please provide re-reference key mouse and human studies.

13. Lines 463-468: Run-on sentence.

14. Lines 490-491: This sentence “As is apparent...” makes it sound like it's the best model to study bacterial evolution in the gut. *E. coli* is a minor player in the human gut microbiome in terms of abundance but has been utilized extensively in molecular biology, microbiology, etc. because of its tractability and relevance to some human diseases. Considering highlighting that *E. coli* isn't “the” model but “a” model.

15. Line 533: Please consider changing “superior” to “more fit”

16. Lines 534, 536: Consider mentioning days, weeks, months, decades, centuries, millennia and how these time scales are currently considered (or not considered) by the microbiome field.

Referee: 3

Comments to the Author(s)

This is a very nice and well written review on gut microbial evolution with a focus on discussing the importance of ecological opportunity. The author carefully cites most of the recent and older literature relevant to the subject and provides new ways forward.

I very much appreciated reading this paper.

I have a couple of really minor points that the author may consider.

line 93 can increase, instead of increase can

line 124 correct referencing.

line 136-154 if possible, shorten and give a more precise, and thus more testable, definition.

line 247: streptomycin treat lab mice in theory and in practice maintain a lot of anaerobic species.

line 447: consider adding evidence from mice provided in Frazão, Sousa, Lassig, Gordo 2019 PNAS.

Author's Response to Decision Letter for (RSPB-2019-1964.R0)

See Appendix A.

Decision letter (RSPB-2019-1964.R1)

30-Oct-2019

Dear Dr Scanlan

I am pleased to inform you that your manuscript entitled "Microbial evolution and ecological opportunity in the gut environment" has been accepted for publication in Proceedings B.

Open Access

Paper charges

Sincerely,

Professor Hans Heesterbeek

Associate Editor:

Board Member

Comments to Author:

Thank you again for your excellent contribution to the special issue, and for so carefully taking into account the comments from the three reviewers. The revised manuscript has satisfactorily addressed all issues raised, and we look forward to including it in the issue!

Appendix A

University College Cork,
17/10/2019

Dear Editors and Reviewers,

Thank you very much to the editor and all three reviewers for their positive responses and excellent feedback. All comments are very constructive and the provision of additional ideas and references to include are much appreciated. I took on board the vast majority of suggestions and introduced several new paragraphs to the original text (see tracked changes in manuscript below). However, this resulted in the manuscript exceeding the 8500 word (10 page limit) by almost 700 words and this revision got unsubmitted. Based on this I had to go back to the text and remove some of the sections that I had added in based on reviewers comments as well as revising some of the original content including removing references. I am sorry that I cannot include some of the suggested comments but it would have required sacrificing some of the original script which, based on your comments, all of you appeared largely happy with. I have tried to address most of your comments in the text where possible but based on the page restrictions but I had to make some hard decisions as to what include/exclude. Please see my responses in italics below with appropriate references to the text where necessary.

Thank you again for the time taken to review my work.

*All the very best,
Pauline Scanlan*

Reviewer(s)' Comments to Author:

Referee: 1

Comments to the Author(s)

I very much enjoyed reading this review, and found it struck a good balance between detailed examples and big picture ideas. In my mind, there are a couple other ideas that would be important to add in the context of microbial evolution and biological opportunity.

1. A brief discussion of biogeography, and how biotic and abiotic niches within the gut provide opportunities for evolution and ecological opportunity. As an addition to this, sequencing the stool provides only a time and space-averaged perspective.

- Mark Welch, J. L., Rossetti, B. J., Rieken, C. W., Dewhirst, F. E. & Borisy, G. G. Bio-

geography of a human oral microbiome at the micron scale. Proc. Natl. Acad. Sci. 201522149 (2016). doi:10.1073/pnas.1522149113

- Lavelle, A., Lennon, G., Winter, D. C. & O'Connell, P. R. Colonic biogeography in health and ulcerative colitis. Gut Microbes 7, 435–442 (2016).
- Tropini, C., Earle, K. A., Huang, K. C. & Sonnenburg, J. L. The Gut Microbiome: Connecting Spatial Organization to Function. Cell Host and Microbe 21, 433–442 (2017).
- Donaldson, G. P., Lee, S. M. & Mazmanian, S. K. Gut biogeography of the bacterial microbiota. Nat. Rev. Microbiol. 14, 20–32 (2016).

Thank you for the suggestion, I have now included a section of biogeography together with associated references in the text, please see lines 312-320.

2. A brief discussion of how metagenomics can also be used for determining factors that are essential in a quantitative understanding of evolution, such as determination of generation time through techniques such as PTR or iRep.

- Brown, C. T., Olm, M. R., Thomas, B. C. & Banfield, J. F. Measurement of bacterial replication rates in microbial communities. Nat. Publ. Gr. (2016). doi:10.1038/nbt.3704
- Korem, T., Zeevi, D., Suez, J., Weinberger, A., Avnit-Sagi, T., Pompan-Lotan, M., Matot, E., Jona, G., Harmelin, A., Cohen, N., Sirota-Madi, A., Thaïss, C. A., Pevsner-Fischer, M., Sorek, R., Xavier, R. J., Elinav, E. & Segal, E. Growth dynamics of gut microbiota in health and disease inferred from single metagenomic samples. Science (80-.). 349, 1101–1106 (2015).

I had included a lengthy section this but had to remove it due to word limitations.

Minor comments:

Line 73: It is worth also noting that physical sampling may also mask the real bacterial diversity due to extraction of DNA from a few hundred micrograms of material from a total of hundreds of grams, which may miss bacteria coming from rare biogeographical niches such as crypts, tight mucus etc.

Another issue is the use of very small quantity of faeces (~milligrams) as sample material for DNA extraction which could limit the detection of rare species and species that are present in biogeographical niches such as crypts or the mucosa.

I agree that current sampling approaches do impact on estimates of diversity but the key point to this paragraph is that if you are interested in gathering information on microbial evolution and diversity below the species level then it is the limited genetic resolution of e.g. 16S rRNA is what primarily affects our understanding of this level of diversity... so even if we applied 16S analysis to a larger volume of stool or a sample from another site (e.g. mucosa) that could in principle contain these rare species they would still only be profiled at a genus level or higher.

Line 170: I think here it should be highlighted that environmental exposure plays a major role in seeding the neonatal microbiota, and that only a few vertically transmitted species appeared to be maintained over the long term.

- Korpela, K., Costea, P., Coelho, L. P., Kandels-Lewis, S., Willemsen, G., Boomsma, D. I., Segata, N. & Bork, P. Selective maternal seeding and environment shape the human gut microbiome. *Genome Res.* (2018). doi:10.1101/gr.233940.117
- Donald, B. and McCoy, K. D. Maternal microbiota in pregnancy and early life. *Science.* (2019). doi: 10.1126/science.aay0618

I do agree that in addition to the maternal microbiota the environment is important to seeding the microbiome (particularly in C-section cases) and I have included a sentence to say that both the maternal microbiota and environmental exposure play the primary role in seeding the neonate microbiota with associated references. See starting Line 169 "For example, the neonate gut can be seeded by microbes from range of sources both maternal (birth canal, faeces, skin, breastmilk) and environmental (hospital setting, home environment, other humans,)..."

With regard to strain persistence, I have included another reference in relation to this (Nayfach, S., et al., An integrated metagenomics pipeline for strain profiling reveals novel

*patterns of bacterial transmission and biogeography. *Genome Res*, 2016. **26**(11): p. 1612-*

1625.) but it is unclear to me based on the current literature to what extent vertically transmitted species are maintained in the long-term i.e. beyond one year. Whilst metagenomics studies have suggested that bacterial strains do get replaced over time there may be alternate explanations i.e. maternally acquired strains may still be present but as the gut microbiome matures other microbes come to predominate and these initial microbes might be below thresholds of detection owing to issues with faecal sampling you rightly outlined in an earlier comment and/or it may also be that strains have diverged sufficiently such that metagenomics controls and cut-offs result in them getting binned as a replacement rather than an evolved strain that originated from the mother. Given the current lack of solid evidence and general lack of longitudinal studies with sufficient resolution to show that strains are not maintained I think it is a little bit early in this area of research to indicate that they are replaced in the long-term.

Line 380: Over the counter drugs are likely to have as big of an impact as antibiotics: antacids, laxatives, anti-inflammatory drugs are wildly prevalent and will heavily impact the gut environment.

Yes, I fully agree that this is very important point and I had alluded to "other medications" in Line 378 in reference to the excellent paper by Maier et al "However, the

evolutionary effects of such changes on the gut microbiota remain unknown but disturbance events including exposure to lethal selection pressures such as antibiotics [1] and other medications [2],". It is worth highlighting this point in more detail and I have to the paragraph and included the additional references you have provided, see text lines 395-403.

Line 476: A microcosm could also be encapsulated by a gut on a chip setup.

- Jalili-Firoozinezhad, S., Gazzaniga, F. S., Calamari, E. L., Camacho, D. M., Fadel, C. W., Bein, A., Swenor, B., Nestor, B., Cronce, M. J., Tovaglieri, A., Levy, O., Gregory, K. E., Breault, D. T., Cabral, J. M. S., Kasper, D. L., Novak, R. & Ingber, D. E. A complex human gut microbiome cultured in an anaerobic intestine-on-a-chip. *Nat. Biomed. Eng.* 3, 520–531 (2019).

Thank you for providing this reference, I was unaware of this paper and it was great to read more about this developing technology. I think it could be very useful to the field in the future if it was easy to set up, run for extended periods and economical to run. I have included a section on it see lines 489-498.

Referee: 2

Comments to the Author(s)

The author provides a thoughtful perspective on current research in the gut microbiome field and how current and future work may benefit from a more thorough consideration of ecological and evolutionary principles. This manuscript highlights opportunities to unite ecological/evolutionary theory, microbiome science, and bacterial genetics and is likely to be of interest to investigators who take all three of these foci in their work. This is an important perspective that should be adopted broadly in the microbiome field.

Broadly, this article is well written and it highlights many key studies from an eco/evolutionary vantage point. However, I feel that the manuscript would benefit from several revisions. These suggested revisions are highlighted below and mainly relate to delineating interpretations made by the authors of the primary literature versus interpretations made by the author of the review. Additional concerns relating to word choice, organization, and figure layout are also discussed.

Major comments

1. Line 194-196: The conclusion "...suggests that although this strain was successfully transmitted from mother to infant it continued to evolve even over the short period of three months" was not a key conclusion from Asnicar et al. (2017) *mSystems*. Though the authors do show data that demonstrates some divergence, this is an interpretation of the author of the manuscript in review. Clearer delineations throughout the manuscript of interpretations from the author and those already made in the

primary literature will be important. This re-framing will be essential, in my opinion, to support the author's assertion from lines 551-553 that "microbiome researchers are largely unaware of the capacity for microbes to evolve in real-time and have overlooked the potential importance of evolution to their research."

As I wrote this paper I tried to be very careful in my wording and choice of language to highlight that much of the ideas and interpretations are mine and are speculative by using terms such as this might suggest, "This data may indicate" "In line with this reasoning" etc.... I clearly state in text that there are no explicit studies of ecological opportunity in the gut line 177 and I deliberately take care to not make any firm conclusions in the text with respect to content from other papers i.e with respect to the Aniscar paper I do not explicitly conclude or state that this is a key conclusion the paper (which I agree it isn't and the authors don't even discuss these data further) but what I have tried to do is suggest that this is one possible explanation i.e. "This data may indicate...".... I then follow up this speculative point with another reference that does explicitly address this possibility in the text i.e. I think it is therefore very clear to the reader which is my mere speculation and which is something that the authors of the original publication have speculated on.

When I wrote this paper I actually had the naïve reader in mind i.e. microbiome researchers that do not work on microbial ecology or evolution (I outlined this in my letter to the editors upon submission) and from my experience most microbiome scientists are very much unaware of the capacity for rapid microbial evolution and don't even think about it in their research and when designing and interpreting results from experiments. I am very glad to read that you think this will be important to those working in ecological/evolutionary theory, microbiome science, and bacterial genetics but I think for those working in eco/evo I am preaching to the converted and therefore my aim was to spark the imagination of those that work in microbiome science but don't factor in the importance of evolution in their work.

2. The substance of the article speaks for itself and manuscript's only figure doesn't positively impact conclusions drawn from the text (even for a visual learner like myself). The author could consider reorganizing this figure into multiple figures that follow the text more closely, building a more informative figure, or including no figure at all.

I am sorry to hear this, I am a visual learner too but I hope that the inclusion of this figure hopefully makes more sense now that I have outlined what my primary aim for a target audience is in the response above. Most researchers that do not explicitly work in the field of experimental evolution and ecology do not know what the basic experimental design and approach is when wishing to address a hypothesis/perform an experiment in this area. I think this figure highlights how an integrated approach whereby people combine experimental evolution models, culturomics, sequencing of natural

occurring isolates and metagenomics is the best way forward. With this integrated approach in mind fragmenting this figure and linking it to the text breaks up a key message that an integrated approach is the best way forward. However, if this figure still does not appeal to you with the new information of my target audience in mind I am happy to remove it. And now that I have a page limitation I think I may not have the space to include.

3. Lines 281-285 and lines 496-499: This is a reasonable interpretation but an alternative interpretation is that there are differences observed in the mouse experiments because the mice were inoculated with an exogenous strain rather than what was done in the human studies, where an already entrenched strain were used. If there were a fundamental difference in evolution and ecological opportunity between "mouse E. coli" versus "human E. coli," an experiment observing native E. coli in mice or a non-native E. coli inoculated into humans would be essential. *I totally agree and I have said as much throughout the text – the reason we probably observe such rapid evolution and adaption in mouse models is because we are using non-native/non murine strainsI explicitly state that we might learn a lot more about ecological opportunity if were to use "murine strains to colonise murine models" in the following section*

Line 279-288 "As outlined this in stark contrast to what has been observed in many of the aforementioned experimental evolution studies using E. coli. However, given that the strain sequenced was dominant within the E. coli population of that individual this would suggest that perhaps it is already well adapted to its host and is therefore evolving neutrally through time [3]. With respect to this finding the application of an experimental evolution approach using murine strains to colonise murine models may provide a more meaningful insight into ecological opportunity and bacterial adaption to the gut environment relative to what has been found in aforementioned mouse models studies that have used laboratory or human strains."

I also clearly outline the importance of niche discordance to observed results see, Line 267-271 "However, variation observed across these different studies also indicate that the genotype of the colonising strain and the degree of niche discordance between it and the environment encountered is also of critical importance to understanding the likely targets of selection and the underlying genetics of adaptation associated with colonisation of the gut environment".

I also outline in the text that niche discordance is likely higher for non-murine strains e.g.

Line 223 -226 "The E. coli strain used was a laboratory strain and not isolated from the murine gut and therefore one could anticipate that niche discordance is certainly higher for this laboratory strain than a native murine E. coli strain".

In lines 281-283 I outline that perhaps the reason the author didn't detect evidence for adaptive evolution was because it was a dominant (e.g. entrenched strain) However, given that the strain sequenced was dominant within the E. coli population of that in-

dividual this might indicate that perhaps it is already well adapted to its host and is therefore evolving neutrally through time [3].

4. Lines 369, 422: In addition to recent decades, the author should consider a brief discussion of changes to the human microbiome over longer time scales to frame the novelty of their viewpoint that evolution and ecological opportunity are operating at more recent timescales.

Although this is a very important point there is very little information in the literature about microbiome evolution over varying and longer time-scales and I think that tackling this question probably warrants another paper with an extensive discussion on the considerable differences in microbiomes we see in Western versus non-Western populations and what this can tell us about selection pressures impacting on the microbiome. However, despite the lack of literature on this it has been shown that diet is likely the main driver of evolution over the course of human history and I have highlighted this point in the text with associated references e.g. Lines 119-121. " Several studies have used (meta)genomic comparative approaches to show that adaption to host diet has played a primary role in shaping the microbial diversity of the human gut over the course of human evolutionary history [4, 5]."

5. Lines 549-561: I feel that the author overlooked an important opportunity to discuss and conceptual roadblocks that existed previously (e.g. a decade ago) which do not exist now. Once you have the theory in mind, how do you leverage technology to design the appropriate experiment? We can't fault previous work if the methods did not exist to suitably address a hypothesis. That is, awareness of evolutionary theory is only the beginning and technological hurdles, rather than ignorance, may have fueled the microbiome field's lack of incorporation of eco/evo.

I do agree in part... but the vast majority of technologies required to perform an experimental evolution using gut microbes all existed a decade ago (and some long-before) e.g. Chemostats, microcosms, germ-free mouse models, anaerobic culturing methodologies, mutant libraries, capacity to genetically engineer strains etc but perhaps what did not exist was access to low cost sequencing. Based on the vast majority of my interactions with microbiome scientists (from immunological, medical, microbiology etc backgrounds) both at home and internationally at conferences (with exceptions of course!... and these are usually at specific evolution sessions...), I have found that it is not technological roadblocks that have held back the study of microbial evolution in the gut but as I have tried to highlight in lines starting 526 that the lack of research to date is likely a disconnect between microbiome science and evolution and it is largely borne out of unawareness (ignorance for want of a better word), lack of exposure to evolutionary studies etc. The study of microbial evolution in the gut is nascent and in my experience those working in the area are evolutionary biologists first and microbiome scientists second not the other way around. I do agree with you that it is very im-

portant to point out that technology does play a role in limiting the scope of studies and in line with your comment I have included the following sentence in the text "However, the study of evolution in complex communities is not trivial and technological roadblocks (e.g. low-cost, high-through put sequencing is only available in the last decade) may have also stymied progress." I also think that studying evolution in a complex ecosystem like the gut is extremely challenging which I have alluded to in the text elsewhere.

Minor comments

1. Line 70: Please be more specific "bacterial diversity at the species level or higher" should be "...at the species level or at higher taxonomic resolution."

Thank you, changed

2. Line 90: "drift" not "draft"

I was referring to both genetic drift and genetic draft in this sentence

3. Line 156: It may be helpful to the reader to highlight longer time scales after the shorter time scales are discussed (e.g. the text from later in the review [lines 329-342] about work done on adaptive radiation in the CF lung).

*I am not sure if I fully understand this suggestion – does the reviewer suggest I move this section up to line 156? I do feel that the section on CF is appropriate where it is as it is included in a section relating to evolution with the human body and even if some of the studies highlighted are over longer time-scales (years as opposed to days, weeks, months) I think that this distinction is apparent in the text. I have included further information in the text on the time-scales of the *B. dolosa* expt.*

4. Lines 156-158: Please provide references.

Thanks you, references have now been added

5. Line 181: "Inaccessible to the host," consider re-wording to "inaccessible to host metabolism."

*Thank you, I have reworded this sentence to " Many infant-associated *Bifidobacterium* species (e.g. *Bifidobacterium breve*, *Bifidobacterium bifidum*) can utilise specific glycans (human milk oligosaccharides) found in breastmilk that cannot be metabolised by the host [6, 7]."*

6. Line 184: "Get filtered out more readily" is imprecise. Consider changing to "ex-

cluded from the gut" or "outcompeted by other microbes and cleared from the gut environment."

Changed as recommended

7. Line 193: "they had a degree of nucleotide divergence." Was this degree of divergence significant?

This was not clear from the paper if the degree of nucleotide divergence was significant statistically and/or biologically....but this is a complex (bordering philosophical) area e.g. a single SNP can confer antibiotic resistance... this is a biologically relevant phenotype but may not be statistically significant if you are using certain genome similarity % cut-off thresholds for species designation etc.

8. Line 236-239. What were the strains that emerged?

The ancestral strain used in this experiments was K12 MG1655 and three variants of this strain emerged over the course of the experiments. I describe the genetic changes associated with these phenotypes in the text.

9. Line 245: Please refer to K12 MG1655 as in lines 214 and 235. Or choose another consistent strain name in all cases.

Changed to K12 MG1655

10. Lines 299-302: Run-on sentence.

Changed.....

11. Line 303: The body also undergoes immunological changes throughout an individual's lifetime.

- I have added immunological into this sentence ... "it undergoes considerable physical, immunological and physiological change".... and changes in host immunology are also mentioned in Line 326 ".....maturation and function of the host immune and endocrine systems....."

12. Lines 455, 457: Please provide re-reference key mouse and human studies.

These have been added

13. Lines 463-468: Run-on sentence.

Changed

14. Lines 490-491: This sentence "As is apparent..." makes it sound like it's the best model to study bacterial evolution in the gut. *E. coli* is a minor player in the human gut microbiome in terms of abundance but has been utilized extensively in molecular biology, microbiology, etc. because of its tractability and relevance to some human diseases. Considering highlighting that *E. coli* isn't "the" model but "a" model.

Thank you, this was not the intention to have E. coli to be lauded as the best but rather that most widely used/studied etc.... I have changed the text as follows "As is apparent from the text, E. coli is the most widely used model bacterium to study bacterial evolution in the gut."

15. Line 533: Please consider changing "superior" to "more fit"

Changed to "...or being replaced by a strain that is fitter..."

16. Lines 534, 536: Consider mentioning days, weeks, months, decades, centuries, millennia and how these time scales are currently considered (or not considered) by the microbiome field.

With reference to point 4 in your earlier comment I agree that delineating scales is hugely relevant given that time-scales are largely not considered in the microbiome field with some exceptions such as pre and post industrialisation and major shifts in human existence e.g. adoption of agriculture etc. However, I think that this warrants a separate perspective to be give it the attention and rigour it deserves.

Referee: 3

Comments to the Author(s)

This is a very nice and well written review on gut microbial evolution with a focus on discussing the importance of ecological opportunity. The author carefully cites most of the recent and older literature relevant to the subject and provides new ways forward.

I very much appreciated reading this paper. *(Thank you!)*

I have a couple of really minor points that the author may consider.

line 93 can increase, instead of increase can

Changed

line 124 correct referencing.

Changed

line 136-154 if possible, shorten and give a more precise, and thus more testable, definition.

Unfortunately, the definition is a little lengthy but this is necessary to include the two relevant aspects of ecological opportunity (i.e. niche availability and niche discordance) as per the recent publication. The central focus of the 2015 review that I cite for the definition was that ecological opportunity be properly defined and consolidated into a definition such that it included all relevant components rather than have it imprecise and fragmented as it was in the literature up until 2015.

line 247: streptomycin treat lab mice in theory and in practice maintain a lot of anaerobic species.

Thank you, I have changed the sentence to include this point.

line 447: consider adding evidence from mice provided in Frazão, Sousa, Lassig, Gordo 2019 PNAS.

This is a really nice paper and I have included a reference to it in the text, Line 476. I don't have the space to explicitly address the relative importance of HGT compared to de novo mutation with respect to adaptation... however, this is something that is hugely important and I think about it quite a lot but similar to the earlier point about time-scales the role of HGT in adaptation in complex ecosystems such as the gut warrants another review.